# Clinical and laboratory profiles of Oropouche virus disease from the 2024 outbreak in Manaus, Brazilian Amazon

Maria Paula Gomes Mourão[1,2], Gisely Cardoso de Melo[1,2], Joabi Nascimento[1,2], Victor Irungu Mwangi[1,2], Livia Sacchetto[3], Luiz Gustavo Gardinassi[4], Rebeca Linhares Abreu Netto[1,2], Jady Mota[1,2], Sergio Damasceno Pinto[1], Mayara Tavares[1], Liz Moreira Cordeiro[1], Barbara Seffair de Castro de Abreu Brasil[5], Dyandra dos Santos Porto[6], Bianca Daniele Silva de Oliveira[1,2], Ana Carolina Shuan Laco[1,2], Lorenna Pereira de Souza[1,2], Karolaine Oliveira Bentes[1,2], Marcia Castilho[1], Carlos Eduardo Morais[1,2], Heline Silva Lira[1], Luís Felipe Alho[1,2], Flor Ernestina Martínez Espinosa[1,7], Vanderson Sampaio[8], Felipe Murta[1,2], Marco Aurélio Sartim [9], Maurício Lacerda Nogueira[3], Marcus Lacerda[1,7], Wuelton Monteiro [1,2]*

1 Diretoria de Ensino e Pesquisa, Fundação de Medicina Tropical Dr. Heitor Vieira Dourado, Manaus, Brazil, 2 Escola Superior de Ciências da Saúde, Universidade do Estado do Amazonas, Manaus, Brazil, 3 Laboratório de Pesquisas em Virologia, Departamento de Doenças Infecciosas e Parasitárias, Faculdade de Medicina de São José do Rio Preto, São José do Rio Preto, Brazil, 4 Escola de Enfermagem de Ribeirão Preto, Universidade de São Paulo, Ribeirão Preto, Brazil, 5 Escola de Saúde Pública de Manaus, Secretaria Municipal de Saúde, Manaus, Brazil, 6 Programa de Residência Médica em Cirurgia, Hospital Adventista de Manaus, Manaus, Brazil, 7 Instituto Leônidas & Maria Deane, Fundação Oswaldo Cruz, Manaus, Brazil, 8 Instituto Todos Pela Saúde, São Paulo, Brazil, 9 Faculdade de Ciências Farmacêuticas, Universidade Federal do Amazonas, Manaus, Brazil

* wueltonmm@gmail.com

## Abstract

### Background

The 2024 Oropouche virus (OROV) outbreak in Brazil raised public health concerns due to its unprecedented rapid spread, high incidence, and potential neurological complications. OROV symptoms overlap with locally endemic arbovirus diseases, like dengue virus (DENV), complicating diagnosis. The study aimed to compare clinical, laboratory, and immunological profiles in OROV and DENV cases, crucial for improving diagnosis and management.

### Methods

This study analyzed 51 OROV and 78 of DENV cases consecutively enrolled in Manaus, Amazonas, Brazil, and monitored for 28 days. OROV diagnosis was performed by real-time PCR (RT-PCR) using serum and urine samples. OROV RT-PCR positive samples were genotyped. A paired Plaque Reduction Neutralization Test (PRNT) was conducted on samples collected at D1 and D28. Patients with a ≥ 4-fold increase in neutralizing antibody titer between D1 and D28 were considered

**Data availability statement:** All data and materials supporting the findings of this manuscript are presented in the paper and/or the supplementary information files. The Oropouche virus sequencing data generated and/or analyzed during the current study are available in the GISAID repository (GISAID.org/EpiArbo), using the accession numbers available in the Supplementary File (File S4). To access and view the sequences, one needs to register to the site (https://gisaid.org/register/), then LOGIN to the GISAID repository. Once you have a login, you'll enter this area Epiarbo>Oropouche , and then click on search. You will then type in the sequence ID, of which it will appear. When you click on it (the ID), you can see the file information and download the sequences. In the case of OROV, it is an access number for the 3 segments.

**Funding:** This study was supported by the Instituto Todos pela Saúde (grant C2024-0190) and from the Fundação de Amparo à Pesquisa do Estado do Amazonas (FAPEAM) via Pro-Estado (Resolução N. 022/2022), to WMM. It also received support from the Conselho Nacional de Desenvolvimento Científico e Tecnológico (CNPq) (311297/2021-8 to LGG; and PDJ postdoctoral scholarship 175855/2023-4 to VIM); Iniciativa Amazônia +10 (Resolução N.023/2022 to MPGM), and Fiocruz/FIOTEC (ID: PRES-009-FIO-22 to ML). Additional support was from the National Institutes of Health (NIH/USA) via Centers for Research in Emerging Infectious Diseases (grant U01AI151807), INCT Dengue Program (grant 465,425/2014-3), INCT Viral Genomic Surveillance and One Health (grant 405,786/2022-0), and from Fundação de Amparo à Pesquisa do Estado de São Paulo (FAPESP) (grant 2022/03,645-1) awarded to MLN. The funders had no role in study design, data collection and analysis, decision to publish, or preparation of the manuscript.

**Competing interests:** The authors have declared that no competing interests exist.

OROV-positive. Clinical manifestations, hematology, biochemistry, and cytokine profiles were analyzed. Statistical analysis included comparison between OROV and DENV patients.

## Results

Genome sequencing of OROV isolates confirmed presence of a previously reported novel reassortment event, consistent with ongoing localized transmission. Urine RT-PCR demonstrated low positivity compared to serum samples. The paired PRNT increased sensitivity in 45%. Clinically, OROV infection was associated with significantly higher frequencies of severe headache, myalgia, arthralgia, and rash compared to DENV infection ($p < 0.001$). Elevated alanine aminotransferase (ALT) levels were also observed in OROV patients ($p < 0.001$). Immunologically, OROV infection induced significantly increased levels of acute-phase CCL11 (eotaxin), CXCL10, IFN-γ, IL-1RA, and IL-10, which declined by day 28, while IL-5 increased during recovery. In contrast, DENV patients exhibited elevated levels of CCL2, G-CSF, and CCL3 in recovery phase.

## Conclusion

OROV symptoms overlap with DENV underscores the need for syndromic diagnostic approach in endemic regions. Continued genomic surveillance and expanded clinical studies are vital to assess long-term consequences. Given OROV's expanding geographic range, targeted public health measures are essential to mitigate future outbreaks and better understand its pathophysiology.

## Author summary

Oropouche virus (OROV) is an emerging mosquito-borne disease increasingly affecting parts of South America, including the Amazon region. Its symptoms—such as fever, headache, joint and muscle pain—are similar to those of other tropical viruses like dengue, making diagnosis and treatment more challenging. In early 2024, Manaus, Brazil, experienced a major outbreak of OROV alongside a regional surge in dengue cases. This study followed patients with either OROV or dengue for 28 days, aiming to compare their clinical, laboratory, and immune profiles. We found that OROV patients experienced more severe headaches, joint pain, and skin rashes than those with dengue. Blood tests showed signs of inflammation and mild liver stress in OROV cases, but no severe disease. Immune system markers also differed between infections, offering clues for differential diagnosis. Genetic sequencing confirmed that the outbreak stemmed from a local strain of the virus that had undergone changes in recent years. Our findings show how OROV behaves differently from dengue, despite their similar presentation, and underscore the importance of distinguishing them for improved diagnosis. This will help prepare and inform public health responses and guide clinical care in future arbovirus outbreaks.

## Background

Arboviral transmission is affected by land-use changes that include deforestation, forest and habitat fragmentation, agricultural development, mining, irrigation, disorderly urbanization, and suburbanization [1–4]. These changes are prominent in the Brazilian Amazon, and have contributed to the proliferation of incidences of arboviruses [5]. The first half of 2024 was characterized by reports of Oropouche virus disease (OROV) outbreak, an arthropod-borne virus transmitted mainly by infected *Culicoides paraensis* midges and *Culex quinquefasciatus* mosquitoes, particularly in the Americas and Caribbean countries, being Brazil the most affected [6–8]. Traditionally, OROV transmission was designated in the Amazon basin since the 1960s [9]. However, more recent reports have identified the disease in other South America countries and in Cuba [10,11], extending beyond the Amazon region of Brazil [9,10]. Additionally, infections have been diagnosed in travelers from Cuba and Brazil in the United States and Europe [7,10].

Following an incubation period of three to ten days post-exposure, OROV infection typically presents with fever accompanied by headache, malaise, myalgia, arthralgia, chills, photophobia, skin rash, and nausea. These symptoms can last for about a week [10,11]. Notably, OROV symptom recurrence has been in approximately 60% of patients, often involving a second episode that is both more severe and longer-lasting [11,12]. Although rare, severe cases may occur and are typically characterized by intensified systemic manifestations. Neurological complications, such as aseptic meningitis, have also been documented, particularly in Brazil [4,11]. The clinical presentation of OROV closely resembles that of other co-circulating arboviruses, such as Dengue (DENV), Zika, and Chikungunya, which further complicates differential diagnosis in endemic regions. For example, during an OROV outbreak in Manaus between 2007 and 2008, many cases were nearly misclassified as dengue due to overlapping clinical features [13,14]. Furthermore, both OROV and DENV have an urban transmission cycle [2–4], thereby facilitating the potential for concurrent outbreaks and the occurrence of OROV/DENV co-infections.

Due to its self-limiting, mild manifestations or clinical features like other prevalent diseases, underreporting and scarcity of documented wild-type OROV infections occur [15,16]. Despite extensive genomic surveillance and viral characterization, there few robust clinical studies addressing OROV infection – particularly prospective studies with high-quality data describing the clinical and laboratory profiles of affected patients [17]. In light of this gap and responding to the spontaneous presentation of individuals with acute febrile syndrome seeking medical care, our study aimed to conduct a clinical and laboratory evaluation of the 2024 OROV outbreak in Manaus, Amazonas State, Brazil. This was carried out in the context of a concomitant DENV outbreak, and included the implementation of a diagnostic routine.

## Methods

### Ethics statement

The study was approved by the ethics committee of the Fundação de Medicina Tropical-Doutor Heitor Vieira Dourado (FMT-HVD) (approval number CAAE: 67595417.0.0000.0005). All participants provided written informed consent.

### Study design

Patients above 5 years of age presenting with acute fever lasting less than 7 days were recruited at the Fundação de Medicina Tropical Doutor Heitor Vieira Dourado (FMT-HVD), a tertiary care center for infectious diseases in Manaus (3°8′S, 60°1′W), from January to March of 2024. After a clinical screening, patients with malaria confirmed by microscopy were not included. At admission (D1), demographics, epidemiological, and clinical information were collected. A clinical characterization was made by collecting days of symptoms, first symptoms, comorbidities, and clinical manifestations at admission. Clinical evaluation included a comprehensive assessment of systemic, mucocutaneous, gastrointestinal, neurological, and respiratory symptoms, including fever, rash, bleeding, headache (with intensity grading), arthralgia, myalgia, retro-orbital pain, and others as detailed in Table 1 of the Results section. Laboratory tests included whole blood counts,

**Table 1. Clinical profile of the patients at inclusion.**

| Variable | OROV (N = 51) | DENV (N = 78) | p-value[1] | Coinfection (N = 5) |
|---|---|---|---|---|
| **Demographics** | | | | |
| Male, % | 22 (43%) | 41 (53%) | 0.30 | 3 (60%) |
| Age, years (SD) | 40.2 (14.8) | 28.5 (10.9) | <0.001 | 42.20 (18.6) |
| **First symptom, %** | | | 0.02 | |
| Fever | 31 (61%) | 64 (82%) | | 3 (60%) |
| Myalgia | 10 (20%) | 5 (6%) | | 2 (40%) |
| Headache | 7 (14%) | 7 (9%) | | 0 (0%) |
| Rash | 1 (2%) | 2 (3%) | | 0 (0%) |
| Arthralgia | 2 (4%) | 0 (0%) | | 0 (0%) |
| **Symptoms on admission, %** | | | | |
| Pruritus | 12 (24%) | 9 (12%) | 0.07 | 0 (0%) |
| Bleeding | 4 (8%) | 1 (1%) | 0.08 | 1 (20%) |
| Epistaxis | 0 (0%) | 1 (1%) | >0.99 | 1 (20%) |
| Headache | 49 (96%) | 70 (90%) | 0.30 | 5 (100%) |
| Headache intensity | | | 0.004 | |
| Moderate | 15 (31%) | 30 (43%) | | 4 (80%) |
| Mild | 13 (27%) | 29 (41%) | | 0 (0%) |
| Severe | 21 (43%) | 11 (16%) | | 1 (20%) |
| Muscle pain | 48 (94%) | 68 (87%) | 0.20 | 5 (100%) |
| Arthralgia | 39 (76%) | 21 (27%) | <0.001 | 4 (80%) |
| Retro-orbital pain | 34 (67%) | 29 (37%) | 0.001 | 1 (20%) |
| Abdominal pain | 16 (31%) | 17 (22%) | 0.20 | 2 (40%) |
| Vomiting | 14 (27%) | 24 (31%) | 0.70 | 1 (20%) |
| Diarrhea | 17 (33%) | 18 (23%) | 0.20 | 2 (40%) |
| Sore throat | 8 (16%) | 14 (18%) | 0.70 | 2 (40%) |
| Running nose | 2 (4%) | 9 (12%) | 0.20 | 1 (20%) |
| Cough | 7 (14%) | 10 (13%) | 0.90 | 1 (20%) |
| Dyspnea | 3 (6%) | 5 (6%) | >0.99 | 1 (20%) |
| Oliguria | 2 (4%) | 1 (1%) | 0.60 | 0 (0%) |
| Dizziness | 20 (39%) | 21 (27%) | 0.14 | 3 (60%) |
| Syncope | 1 (2%) | 1 (1%) | >0.99 | 0 (0%) |
| Altered consciousness | 1 (2%) | 1 (1%) | >0.99 | 0 (0%) |
| Rash | 18 (35%) | 11 (14%) | 0.005 | 1 (20%) |
| Symmetric | 14 (27%) | 11 (14%) | 0.06 | 1 (20%) |
| Palpable injury | 7 (14%) | 6 (8%) | 0.30 | 0 (0%) |
| Head | 12 (24%) | 5 (6%) | 0.005 | 1 (20%) |
| Torso | 15 (29%) | 8 (10%) | 0.005 | 1 (20%) |
| Upper limbs | 16 (31%) | 10 (13%) | 0.01 | 1 (20%) |
| Lower limbs | 12 (24%) | 5 (6%) | 0.005 | 1 (20%) |
| Palms and soles | 0 (0%) | 1 (1%) | >0.99 | 0 (0%) |
| Petechiae | 6 (12%) | 3 (4%) | 0.20 | 0 (0%) |
| Conjunctival congestion | 5 (10%) | 3 (4%) | 0.30 | 1 (20%) |
| Palpebral edema | 6 (12%) | 5 (6%) | 0.30 | 2 (40%) |
| Oropharyngeal changes | 0 (0%) | 3 (4%) | 0.30 | 0 (0%) |
| Gingival bleeding | 2 (4%) | 0 (0%) | 0.20 | 0 (0%) |
| Menometrorrhagia | 1 (2%) | 0 (0%) | 0.40 | 0 (0%) |

*(Continued)*

**Table 1.** (Continued)

| Variable | OROV (N = 51) | DENV (N = 78) | p-value[1] | Coinfection (N = 5) |
|---|---|---|---|---|
| Enanthema | 1 (2%) | 0 (0%) | 0.40 | 0 (0%) |
| Tourniquet test, % | | | | |
| Positive | 3 (7%) | 4 (5%) | 0.70 | 0 (0%) |

[1]Pearson's Chi-squared test; Fisher's exact test.

platelet counts, albumin, creatine phosphokinase, alanine aminotransferase, aspartate aminotransferase, creatinine, and urea. A tourniquet test was also performed.

Each patient was monitored for 28 days at outpatient clinics, with sociodemographic data, clinical-laboratory findings, and blood samples collected at admission (D1) and on day 28 (D28). D1 Blood samples were tested for OROV and DENV, and serum was collected and stored at -80°.

## Laboratory diagnosis of OROV and DENV

Blood samples were subjected to total nucleic acid extraction using the Maxwell Viral Total Nucleic Acid Purification Kit (Promega, USA). OROV diagnosis was confirmed by real-time PCR (RT-PCR) (QuantStudio 5, Applied Biosystems, Thermo Fisher Scientific, USA) using serum and urine samples (S1 File) [18]. Additionally, a Plaque Reduction Neutralization Test (PRNT) technique was conducted on samples collected at D1 and D28. For this, serum samples were inactivated at 56°C for 60 min and subjected to an assay to determine the specific OROV neutralizing antibody titers. Here, the PRNT assays was performed using 6-well plates containing 80,000 Vero cells/well, of which the OROV virus (~300 PFU-) was incubated with different patient serum dilutions (1:20 – 1:640). The OROV virus used here was isolated from a clinical sample in Feb/2024 during the OROV outbreak in the city of Manaus. The plates were then covered with a semi-solid medium (1X DMEM, 1% FBS, 1.5% CMC) and incubated at 37°C in 5% $CO_2$ for six days. After six days, Vero cells were fixed and stained. PRNT titers were expressed according to 80% plaque inhibition (PRNT80), of which titers with PRNT80 < 20 PFU/mL were considered negative and titers greater than 1:20 PFU/mL in the D28 samples were defined as positive [19,20]. Patients with a ≥ 4-fold increase in neutralizing antibody titer between D1 and D28 were considered OROV-positive.

Dengue was diagnosed using the NS1 (Abbott, Chicago, USA), coupled with serological assays using the Panbio Dengue IgM Capture ELISA test (Abbott, Chicago, USA), and molecular tests using the Biomol ZDC kits (Instituto de Biologia Molecular do Paraná, Brazil). The dengue virus serotypes (DENV1–4) were determined using kits provided by the Ministry of Health that included the ZDC molecular assay kit (Bio-Manguinhos, Brazil) and the Biomol ZDC kit (Instituto de Biologia Molecular do Paraná, Brazil). The ZDC kits were also capable of detecting Zika and Chikungunya viruses. Samples testing positive for Zika or Chikungunya infections were subsequently excluded from analysis. All reference viruses are available from the UTMB World Reference Center for Emerging Viruses and Arboviruses (WRCEVA).

## Virus genotyping

To perform complete genome sequencing, a total of 140 µL of serum samples were submitted to total RNA extraction using QIAamp Viral RNA Mini kit (QIAGEN, Germany), according to the manufacturer's instructions and including a negative template control (nuclease-free water) in each extraction batch. A reverse transcription-quantitative PCR (RT-qPCR) was performed using GoTaq 1-Step RT-qPCR System (Promega, USA) and oligos targeting the Oropouche virus S segment gene (S2 File). The RT-qPCR was conducted in the QuantStudio 3 Real-Time PCR System (Thermo Fisher Scientific, USA), and the results were analyzed in QuantStudio 3 software v1.5.1 (Thermo Fisher Scientific, USA). Results were

interpreted as positive if cycle quantification threshold (Ct) values were less than ≤38 [21]. Positive (BeAn19991 RNA) and negative (nuclease-free water) and negative template control were used in the assays.

The cDNA synthesis, OROV genome amplification, and library preparation were performed using a modified version of the Illumina COVIDSeq Test (Illumina, USA), in which the original primers were replaced with a serotype-specific primer panel for OROV, designed by the Brazil-UK Centre for Arbovirus Discovery, Diagnosis, Genomics, and Epidemiology (CADDE). Each sequencing run included a negative (nuclease-free water) and negative template control during cDNA synthesis and amplicon generation. Library quantification was performed using the Qubit dsDNA HS Assay kit on a Qubit 2.0 Fluorometer (Thermo Fisher Scientific, USA). Quality control of the libraries was verified using a High Sensitivity D1000 ScreenTape kit on the 4150 TapeStation system (Agilent Technologies, USA). PhiX Control v3 was used as a quality control and validation for sequencing runs. Pooled libraries were normalized to 4 nM, denatured with 0.2 N NaOH, and sequenced on the Illumina MiSeq System (Illumina, USA), using a MiSeq Reagent Kit v2 (2 x 150 bp cycles) (Illumina, USA) with a paired-end strategy.

Raw demultiplexed read data generated from sequencing amplicons were processed using Cutadapt v.4.6 [22] to filter low-quality reads (minimum Phred score of Q30), low-quality bases, reads with a minimal length of 50 base pairs (bp) and for primer removal. The cleaned paired-end reads were mapped against the OROV reference genomes (segment L, NC_005776.1; segment M, NC_005775.1; and segment S, NC_005777.1) using BWA v.0.7.17 [23,24]. BAM files are then sorted and indexed using SAMtools v.1.10, and the consensus nucleotide sequence was generated using SAMtools v.1.10 and iVar v. 1.3.1 [24,25] (per-base depth of at least 10 mapped reads were used by default).

OROV complete genome sequences with high coverage, date, and country of collection were retrieved from the Gen-Bank database (https://www.ncbi.nlm.nih.gov/nucleotide/). The nucleotide sequences were aligned using MAFFT v.7.526 [26] and manually curated using AliView v.1.28 [23]. The maximum likelihood (ML) phylogenetic trees were inferred using IQ-TREE v.2.3.6 [27] under the best-fit substitution model estimated by the ModelFinder model-selection method implemented in IQ-TREE (8), according to Bayesian information criterion (BIC). The reliability of branching patterns was tested using the ultrafast bootstrap approximation (UFBoot) with 1,000 bootstrap alignments combined with SH-like Approximate Likelihood-ratio test (SH-aLRT) with 1,000 replicates. The time-scaled ML tree and the root-to-tip genetic distances were inferred using TreeTime. The trees were visualized and edited using R statistical software v.4.3.1 [28] using the ggplot2 v.3.4.2, ggtree v.2.0.0, and scales v.1.2.1 packages.

The OROV sequences originating from our study were screened for potential intra-segment recombination using the Recombination Detection Program (RDP) [29] version 5 using seven different detection algorithms (RDP, Geneconv, Bootscan, Maxchi, Chimaera, 3Seq, and LARD) implemented in the RDP5 package, with default settings.

## Immunological markers

Inflammatory cytokine and chemokine were measured at inclusion and at D28 using the commercial Human ProcartaPlex Mix&Match 25-plex Multiplex Immunoassay kit (lot 343711–000, Invitrogen) targeting the following: Eotaxin (CCL11), FGF-2, G-CSF, GM-CSF, IFN-γ, IL-1β,IL-10, IL-12p70, IL-13, IL-15, IL-17A, IL-1RA, IL-2, IL-4, IL-5, IL-6, IL-8, IP-10, MCP-1, MIP-1α, MIP-1β, PDGF-BB, TNF-α, TREM-1, and VEGF-A. All assays were performed according to the manufacturer's protocols using Luminex xMAP technology. Data quantification of the cytokine and chemokine concentrations was conducted using the Invitrogen ProcartaPlex Analysis App (Thermo Fisher Scientific).

## Statistical analysis

Frequencies were compared by Chi-square test, corrected by Fisher's exact test if necessary. D'Agostino & Pearson omnibus normality test was used to test for normality distribution. Unpaired t test was used for parametric and Mann Whitney test was used for non-parametric comparisons between two groups. Correlation analyses were performed with Spearman's rank correlation method. To explore whether patterns of symptom co-occurrence could distinguish OROV

from DNV, we constructed symptom networks. The network of symptoms was generated by calculating the percentage of patients with each symptom represented at the nodes and the edge represent the proportion of patients reporting both symptoms. This analysis aimed to identify distinct syndromic clusters associated with each virus and evaluate whether such patterns might support syndromic differentiation in outbreak settings. Statistical analyses were calculated with R v4.4.0 and the networks were visualized with Cytoscape v3.10.0. Statistical significance was considered if p < 0.05.

## Results

### Clinical and laboratorial characterization

A total of 644 serum samples were tested, and 51 OROV, 78 DENV, and 5 OROV-DENV coinfection cases were included in this study. Of the 51 patients with OROV, 28 were positive by RT-PCR and 23 were positive only by PRNT. Thus, PRNT showing a ≥ 4-fold increase in neutralizing antibody titer from D1 to D28 elevated sensitivity by 45%. A total of 61 urine samples were submitted to OROV RT-PCR, of which 40 had positive and 21 had negative results in RT-PCR using serum. Of these, 8 urine samples were positive, all also positive in serum (20% of positivity). Eighty-one DENV patients were determined to have dengue virus type 2, whilst two had dengue virus type 1.

The OROV patients were significantly older than the DENV patients (40.2 yrs. vs 28.5 yrs.; p < 0.001). At inclusion, OROV patients were significantly more likely to report myalgia (20% vs. 6%) and less likely to report fever (61% vs. 82%) compared to DENV patients (p = 0.02). On admission, several symptoms were notably more frequent among OROV cases than among the DENV patients: severe headache (43% vs. 16%, p = 0.004), arthralgia (76% vs. 27%, p < 0.001), retro-orbital pain (67% vs. 37%, p = 0.001), and rash (35% vs. 14%, p = 0.005). In OROV patients, the rash was more diffuse, affecting several body segments including head, torso, and upper and lower limbs, when compared to DENV (Table 1). Skin rash presentations of OROV and DENV are shown in Fig 1.

To understand how symptom presentation differed between OROV and DENV cases, we did symptom frequencies and co-occurrence networks that further highlighted differences between the two infections (Fig 2). In OROV-infected patients, clinical symptoms exhibited stronger intercorrelations, forming a distinct cluster centered on headache, conjunctival congestion, and rash, in contrast to that observed in DENV cases. The network analysis provides an exploratory, low-assumption visualization of symptom interrelations that may support further hypothesis generation, and early syndromic cues for targeted diagnosis during arboviral outbreaks in endemic regions.

Liver function tests revealed significantly elevated ALT levels in the OROV patients compared to the DENV patients (47 vs 27 U/L) (p < 0.001). Despite being within the normal ALT range, the significant difference could be indicative of a mild subclinical virus-driven hepatic damage by OROV. There were, however, no remarkable differences in other laboratory profiles on admission (Table 2).

One DENV patient had persistent headaches from D1 to D28. One OROV patient had arthralgia and myalgia, persisting from D1 to D28.

Baseline demographics and symptom profiles were compared between the 28 PCR-confirmed and 23 PRNT-confirmed OROV. Overall, symptom distributions were similar across groups, with joint pain (89% vs. 61%; p = 0.02) and palpebral edema (21% vs. 0%; p = 0.03) reported significantly more frequently among PCR-positive patients (Table 3).

### Virus genotyping

The 28 OROV RT-qPCR positive serum samples presented a mean cycle threshold (Ct) value of 29.6 (SD: 6.4; range: 19.2 to 36.8). We sequenced a total of 11 complete genomes of the virus (S3 File). The sequenced genomes had a mean Ct value of 21.0 and genome coverage of 96.3% (L segment = 97.7%, M segment = 99.3%, S segment = 95.1%). The complete sequenced genomes had an average sequencing depth of 2,667x (L segment = 2779x, M segment = 3178x, S segment = 2045x).

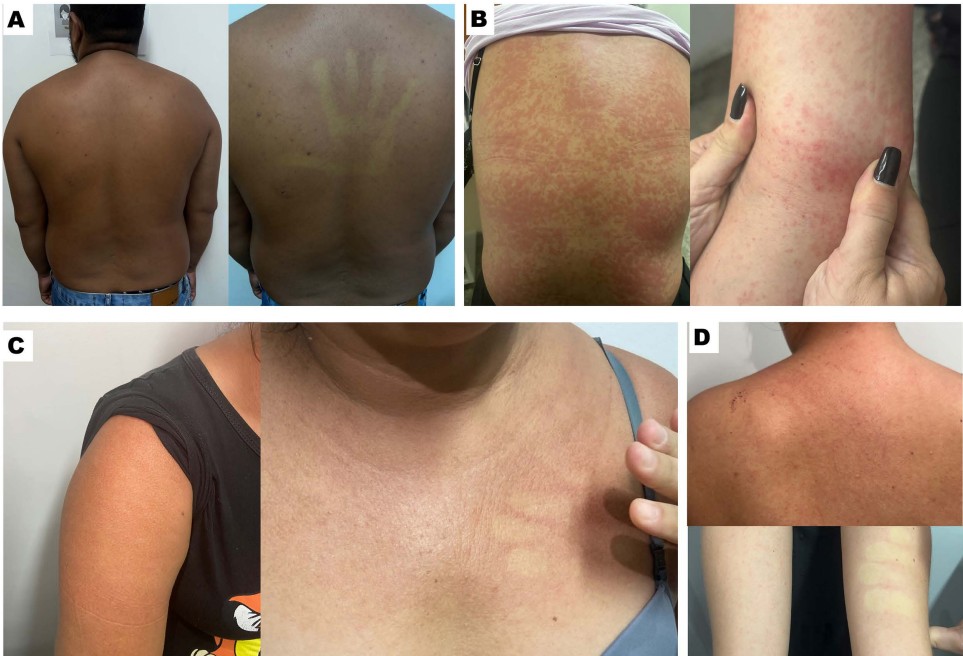

**Fig 1. Skin rash presentations of OROV and DENV infections in the assessed patients. A)** 27-year-old male with 6 days of symptoms diagnosed with OROV presented with rash on torso, upper and lower limbs. Patient also reported fever, arthralgia, myalgia, headache, retro-orbital pain, abdominal pain, vomiting, diarrhea, and dizziness. **B)** 32-year-old female with 6 days of symptoms diagnosed with OROV, presenting with rash on torso, upper and lower limbs. Patient had also reported fever, arthralgia, myalgia, severe headache, pruritus and had a positive tourniquet test. **C)** 43-year-old female with 3 days of symptoms diagnosed with DENV, presenting with rash on torso, upper and lower limbs. Patient had also reported fever, arthralgia, myalgia, pruritus, and mild headache. **D)** 26-year-old female, 8 weeks pregnant and with 3 days of symptoms diagnosed with DENV, presenting with rash on torso, upper and lower limbs with petechiae. Patient had also reported fever, myoarthralgia, mild headache, pruritus, and retro-orbital pain.

Maximum likelihood phylogenetic analysis of our sequenced samples (Fig 3) was constructed for the L (polymerase), M (glycoprotein), and S (nucleoprotein) segments of OROV, incorporating sequences generated in this study alongside reference strains (S4 File). In all three trees, the 2024 Manaus outbreak sequences (white nodes) formed well-supported monophyletic clusters, closely related to 2015–2025 clade previously reported in Brazil collected between 2009 and 2022 (blue nodes), indicating local viral persistence and limited recent diversification (Fig 3). The consistent clustering of the 2024 sequences across all segments suggests limited reassortment and that the outbreak was primarily driven by locally circulating OROV strains as previously reported [6,15].

## Immunological markers

In both DENV and OROV infections, significant changes in immune markers were observed between D1 and D28, but with distinct patterns. In the DENV infection, levels of IL-1RA, IL-10, CXCL10, and CCL2 were significantly higher on D1 compared to D28, while markers like IL-5, G-CSF and CCL3 rose towards D28 (Fig 4). These shifts indicate a dynamic immune response, with notable inflammation at the onset of infection followed by a shift towards a more regulated immune environment at D28.

In contrast, OROV patients showed significantly elevated levels of CCL11 (eotaxin), CXCL10, IFN-γ, IL-1RA and IL-10 at D1, which declined by D28 (all p < 0.05) (Fig 5). Notably, IL-5 was significantly elevated at D28, indicating a potential shift towards the Th2 response later in infection. Worth noting, is that levels of CXCL10 in DENV patients at D28 were higher compared to OROV patients.

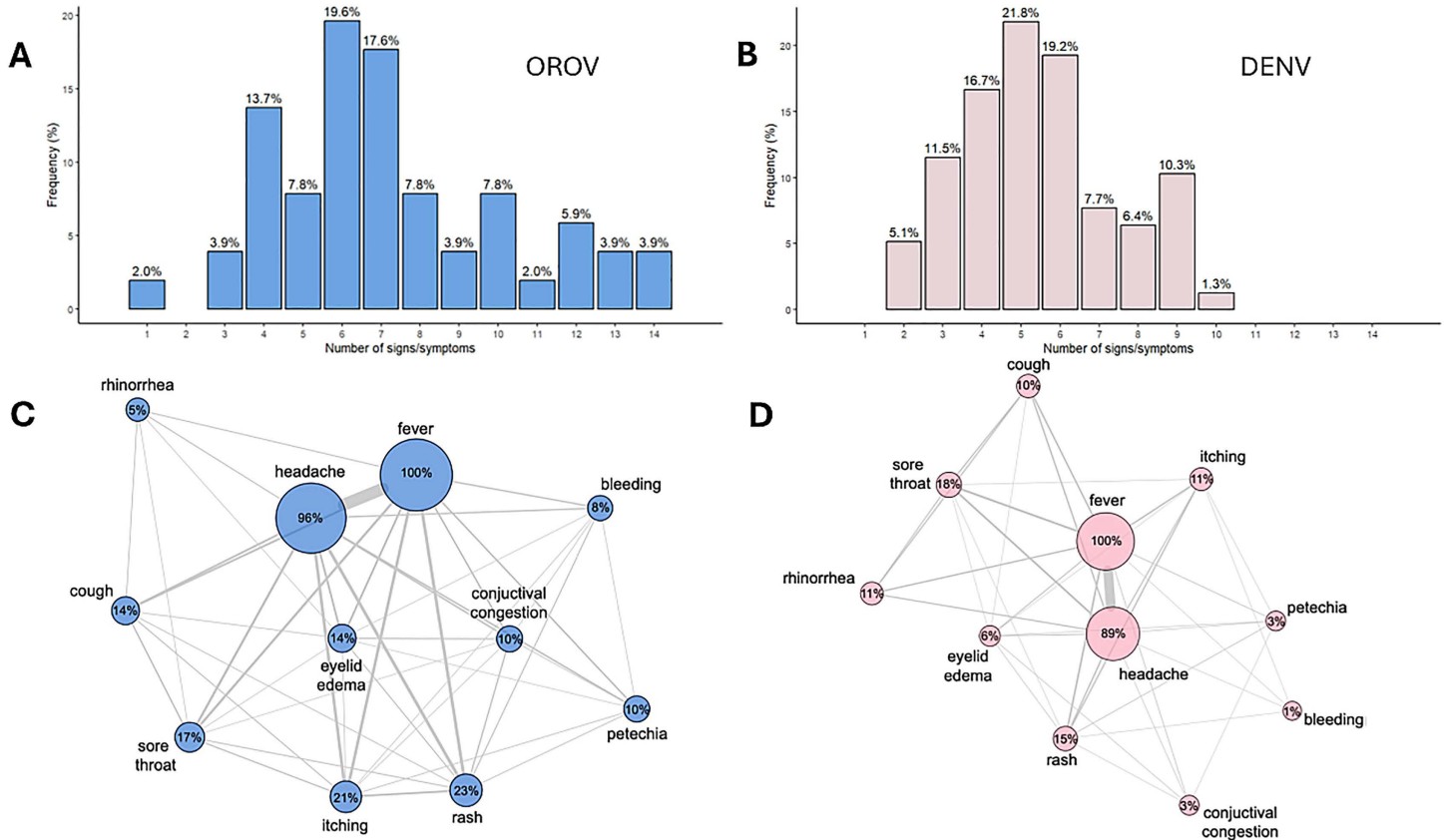

**Fig 2. Symptom burden and co-occurrence patterns in patients with Oropouche virus (OROV) and Dengue virus (DENV) infections.** A) Number of signs and symptoms in OROV patients, B) Number of signs and symptoms in DENV patients, **C)** and D) A network graph showing the co-occurrences of reported symptoms.

## Discussion

OROV and DENV reveal substantially similarities in their epidemiology, clinical manifestations, and impact on public health, presenting unique challenges for diagnosis and management. In the present study, we studied the clinical and laboratorial profiles, including immune markers, of Brazilian patients with OROV or DENV during an unprecedented outbreak of the first disease in Manaus, western Brazilian Amazon in late 2024.

Combining RT-PCR coupled with PRNT magnifies diagnostic sensitivity and is critical in improving case detection for necessary clinical follow-up or epidemiological interventions or investigations. Detection of viral RNA is dependent on viral load and the presence of virus in the sample, however the possibility of low levels of viremia or negative RT-PCR results does not exclude infection [16]. The PRNT test presents a means to increase diagnostic power, as it helps amplifies detection range shows exposure even in the absence of the virus or undetectable viremia [30].

A comparative analysis of the lab parameters and clinical manifestations in active OROV infection (PCR+) relative to the cleared viremia (the PRNT+, but PCR-) suggests that high frequency of joint pain and palpebral edema symptoms may be associated with presence of OROV in circulation. This may be indicative of pathophysiological effects of active infection which mirror in clinical and laboratory manifestations [17]. Likewise the thrombocytopenia and leukopenia are characteristic indicators of viral infections in which the virus impact on the immune system, at times even hiding in

**Table 2. Clinical laboratory profile of the patients at inclusion.**

| Variable | OROV (N = 51) | DENV (N = 78) | p-value[1] | Coinfection (N = 5) |
|---|---|---|---|---|
| Red blood cells (x10$^6$/mm$^3$, IQR) | 4.9 (4.6, 5.4) | 4.8 (4.6, 5.1) | 0.20 | 4.9 (4.7, 5.0) |
| Platelets (x10$^3$/mm$^3$, IQR) | 195.3 (147.2, 234.0) | 189.5 (155.4, 224.2) | 0.80 | 140.6 (129.5, 223.1) |
| MCV (fL, IQR) | 86.8 (84.9, 88.9) | 87.6 (85.3, 90.5) | 0.10 | 90.5 (89.6, 91.5) |
| MCH (fL, IQR) | 28.79 (27.7, 29.9) | 29.3 (28.5, 30.5) | 0.07 | 30.5 (30.1, 31.1) |
| MCHC (fL, IQR) | 33.3 (32.6, 33.8) | 33.5 (32.7, 34.1) | 0.30 | 33.3 (33.2, 34.7) |
| RDW (fL, IQR) | 11.5 (11.1, 12.3) | 11.8 (11.4, 12.6) | 0.30 | 11.3 (11.3, 11.5) |
| MPV (fL, IQR) | 7.3 (6.4, 7.8) | 6.9 (6.3, 7.7) | 0.40 | 6.9 (6.8, 8.5) |
| Leukocytes (x10$^3$/mm$^3$, IQR) | 5.7 (3.7, 8.3) | 5.6 (3.9, 6.9) | 0.60 | 10.4 (6.2, 11.0) |
| Lymphocytes (x10$^3$/mm$^3$, IQR) | 1.5 (0.6-2.8) | 1.5 (0.7-2.3) | >0.99 | 1.2 (0.5-1.5) |
| Monocytes (x10$^3$/mm$^3$, IQR) | 0.5 (0.3-1.0) | 0.6 (0.3, 0.8) | 0.70 | 0.7 (0.3-1.0) |
| Neutrophils (x10$^3$/mm$^3$, IQR) | 3.6 (2.0, 6.1) | 3.5 (2.1-5.0) | 0.80 | 8.4 (4.8-9.1) |
| Albumin (g/dL, IQR) | 4.8 (4.5, 4.9) | 4.8 (4.6, 5.0) | 0.30 | 4.7 (4.4, 5.1) |
| Creatine phosphokinase (U/L; IQR) | 93 (70, 129) | 103 (75, 167) | 0.30 | 92 (71, 97) |
| Aspartate aminotransferase (U/L; IQR) | 34 (23, 62) | 27 (23, 39) | 0.12 | 24 (22, 27) |
| Alanine aminotransferase (U/L; IQR) | 47 (32, 81) | 27 (19, 39) | <0.001 | 22 (20, 46) |
| Creatinine (mg/dL; IQR) | 0.8 (0.7-1.0) | 0.9 (0.7, 1.1) | 0.40 | 1.0 (0.9-1.1) |

[1]Wilcoxon rank sum test.

leukocytes [31]; high monocytes, MPV, CPK, AST, and ALT levels too indicate acute diseases or peak viraemia, with a possibility of hepatic involvement [32].

In recent years, increased travel and population movements have resulted in imported viral infections leading to secondary local transmissions and subsequent outbreaks in susceptible populations. With this in mind, our mutation analysis of the complete OROV genomes obtained from our patients revealed that the by OROV virus lineage responsible for the outbreak was from a previously reported reassortment event [6,7]. With an > 96% shared identity of the viral L, M and S segments with the reference sequences, a silent localized urban transmission of the OROV was ascertained with no spillover from the neighboring countries in the region. Nevertheless, authors theorized a viral evolution (novel reassortment) in the older milder resident clade was responsible for the remarkably infectious OROV epidemic in western Brazil in 2024 [8]. This was subsequently verified by a recent study by Scachetti et al. [15], in which the 2024 epidemic isolate - from Manaus and Coari municipalities of Amazonas state, Brazil - was compared with an older historical isolate (BeAn19991). The 2024 reassortant strain, similar to ours, was concluded to be more virulent. This virulence may also explain the observed high incidence of Oropouche in 2024, potentially driven by the reassortant strain's high replication rate and a low neutralizing capacity within the general in the population [15]. Other factors promoting an explosion of vector population, geographic spread of arboviral diseases into human populations due to vector expansion, and climate change, may have also contributed to the observed concurrent DENV and OROV epidemic [33].

Regarding clinical presentation in the two infections, patients analyzed in this study exhibited classic clinical manifestations of arboviral infections [34]. Our data demonstrate that both infections can be quite similar. However, individuals with OROV infection presented with a higher frequency of initial symptoms at the time of admission, possibly due to the virulent nature of the reassorting OROV established to be in circulation.

A systematic review from 15 studies involving 806 OROV-infected patients demonstrated a highly similar profile, of which fever, headache, myalgia, arthralgia, and retro-orbital pain were the most frequently reported symptoms [17]. Headaches were notably more intense among OROV-infected patients compared to those with DENV infection, potentially reflecting mild neurological involvement characteristic of OROV infection. While correlation does not imply causation,

**Table 3. Comparison of baseline symptoms in PCR- and PRNT-confirmed OROV cases.**

| Variable | Diagnosis | | p-value[2] |
|---|---|---|---|
| | **PCR**, N = 28[1] | **PRNT**, N = 23[1] | |
| Sex, males (%) | 14 (50%) | 8 (35%) | 0.3 |
| Age, years (IQR) | 34 (27, 51) | 41 (32, 58) | 0.2 |
| First symptom | | | 0.3 |
| Fever | 18 (64%) | 13 (57%) | |
| Headache | 3 (11%) | 7 (30%) | |
| Myalgia | 5 (18%) | 2 (8.7%) | |
| Arthralgia | 1 (3.6%) | 1 (4.3%) | |
| Rash | 1 (3.6%) | 0 (0%) | |
| Pruritus | 5 (18%) | 7 (30%) | 0.3 |
| Rash | 7 (25%) | 5 (22%) | 0.8 |
| Bleeding | 3 (11%) | 1 (4.3%) | 0.6 |
| Epistaxis | 0 (0%) | 0 (0%) | |
| Headache | 27 (96%) | 22 (96%) | >0.9 |
| Intensity of headache | | | 0.5 |
| Severe | 10 (37%) | 11 (50%) | |
| Mild | 10 (37%) | 5 (23%) | |
| Moderate | 7 (26%) | 6 (27%) | |
| Muscle pain | 28 (100%) | 20 (87%) | 0.08 |
| Joint pain | 25 (89%) | 14 (61%) | 0.02 |
| Retro orbital pain | 19 (68%) | 15 (65%) | 0.8 |
| Abdominal pain | 9 (32%) | 7 (30%) | 0.9 |
| Vomiting | 9 (32%) | 5 (22%) | 0.4 |
| Diarrhea | 8 (29%) | 9 (39%) | 0.4 |
| Sore throat | 3 (11%) | 5 (22%) | 0.4 |
| Runny nose | 1 (3.6%) | 1 (4.3%) | >0.9 |
| Cough | 2 (7.1%) | 5 (22%) | 0.2 |
| Dyspnea | 1 (3.6%) | 2 (8.7%) | 0.6 |
| Oliguria | 0 (0%) | 2 (8.7%) | 0.2 |
| Dizziness | 10 (36%) | 10 (43%) | 0.6 |
| Syncope | 1 (3.6%) | 0 (0%) | >0.9 |
| Altered consciousness | 0 (0%) | 1 (4.3%) | 0.5 |
| Convulsion | 0 (0%) | 0 (0%) | |
| Jaundice | 0 (0%) | 0 (0%) | |
| Rash | 11 (39%) | 7 (30%) | 0.5 |
| Symmetric | 8 (29%) | 6 (26%) | 0.8 |
| Palpable injury | 3 (11%) | 4 (17%) | 0.7 |
| Head | 6 (21%) | 6 (26%) | 0.7 |
| Torso | 8 (29%) | 7 (30%) | 0.9 |
| Upper limbs | 9 (32%) | 7 (30%) | 0.9 |
| Lower limbs | 6 (21%) | 6 (26%) | 0.7 |
| Palms and soles | 0 (0%) | 0 (0%) | |
| Petechiae | 3 (11%) | 3 (13%) | >0.9 |
| Conjunctival congestion | 4 (14%) | 1 (4.3%) | 0.4 |
| Palpebral edema | 6 (21%) | 0 (0%) | 0.03 |
| Menometrorrhagia | 1 (3.6%) | 0 (0%) | >0.9 |

*(Continued)*

**Table 3.** (Continued)

| Variable | Diagnosis | | p-value[2] |
|---|---|---|---|
| | **PCR**, N = 28[1] | **PRNT**, N = 23[1] | |
| Enanthema | 0 (0%) | 1 (4.3%) | 0.5 |
| Tourniquet test | 2 (100%) | 1 (100%) | |

[1]n/ N (%); Median (IQR).

[2]Pearson's Chi-squared test; Wilcoxon rank sum test; Fisher's exact test.

PCR-confirmed individuals exhibited significantly lower median platelet (159,900/ mm$^3$ vs. 234,800/ mm$^3$; $p < 0.001$) and leukocyte counts (4,195/ mm$^3$ vs. 8,271/ mm$^3$; $p < 0.001$) compared to PRNT-confirmed cases. Additionally, PCR-diagnosed patients had higher median values of monocytes (11.0 vs. 7.0; $p = 0.005$), mean platelet volume (MPV) (7.42 fL vs. 6.74 fL; $p = 0.03$), CPK (111 U/L vs. 79 U/L; $p = 0.03$), AST (49 U/L vs. 23 U/L; $p < 0.001$), ALT (61 U/L vs. 34 U/L; $p = 0.03$) (Table 4). No significant differences were observed in other hematological and biochemical parameters.

**Table 4. Hematological and biochemical profile of OROV-positive patients by different diagnostic methods (PCR versus PRNT).**

| Variable | Diagnosis | | |
|---|---|---|---|
| | **PCR**, N = 28[1] | **PRNT**, N = 23[1] | p-value[2] |
| Red blood cells (x10$^6$/mm$^3$, IQR) | 4.79 (4.62, 5.13) | 5.30 (4.57, 5.49) | 0.2 |
| Platelets (x10$^3$/mm$^3$, IQR) | 159.9 (132.38, 195.93) | 234.8 (197.45, 273.1) | <0.001 |
| MCV (fL, IQR) | 87.1 (85.2, 89.4) | 85.6 (83.5, 88.4) | 0.2 |
| MCH (fL, IQR) | 29.02 (27.97, 29.99) | 28.76 (27.75, 29.55) | 0.5 |
| MCHC (fL, IQR) | 33.27 (32.55, 33.61) | 33.38 (32.59, 33.99) | 0.6 |
| RDW (fL, IQR) | 11.36 (11.02, 12.01) | 11.79 (11.22, 12.72) | 0.12 |
| MPV (fL, IQR) | 7.42 (6.86, 7.94) | 6.74 (6.13, 7.44) | 0.03 |
| Leukocytes (x10$^3$/mm$^3$, IQR) | 4.195 (3.307, 5.55) | 8.271 (6.45, 10.706) | <0.001 |
| Lymphocytes (x10$^3$/mm$^3$, IQR) | 20 (16, 34) | 28 (20, 34) | 0.3 |
| Monocytes (x10$^3$/mm$^3$, IQR) | 11.00 (8.50, 13.00) | 7.00 (6.00, 10.00) | 0.005 |
| Neutrophils (x10$^3$/mm$^3$, IQR) | 66 (52, 73) | 59 (56, 69) | 0.7 |
| Albumin (g/dL, IQR) | 4.75 (4.50, 4.80) | 4.80 (4.50, 4.95) | 0.6 |
| Creatine phosphokinase (U/L; IQR) | 111 (81, 150) | 79 (63, 108) | 0.03 |
| Aspartate aminotransferase (U/L; IQR) | 49 (34, 69) | 23 (18, 32) | <0.001 |
| Alanine aminotransferase (U/L; IQR) | 61 (38, 82) | 34 (22, 59) | 0.03 |
| Creatinine (mg/dL; IQR) | 0.85 (0.73, 1.15) | 0.80 (0.70, 1.00) | 0.4 |

[1]Median (IQR).

[2]Wilcoxon rank sum test; Wilcoxon rank sum exact test.

several mechanisms may underlie these symptoms. In arboviral infections, headaches are often linked to complex pathophysiological processes involving to the host's immune response, neuroinflammation, and vascular dysfunction [35]. Most cases of neurological impairment, as described in the literature, attributed to OROV are severe, requiring cerebrospinal fluid puncture and hospitalization. Here, we had a series of non-severe cases followed over time, but they possibly express this frustrating neurological impairment in a population with endemic exposure. With further investigations and the establishment of a standardized scale, headache intensity, when assessed alongside other frequent symptoms like fever, rash, and itching, could serve as a valuable clinical parameter for differentiating OROV and DENV infections, especially during co-circulation. Additionally, other studies have established that arbovirus infection and the accompanying

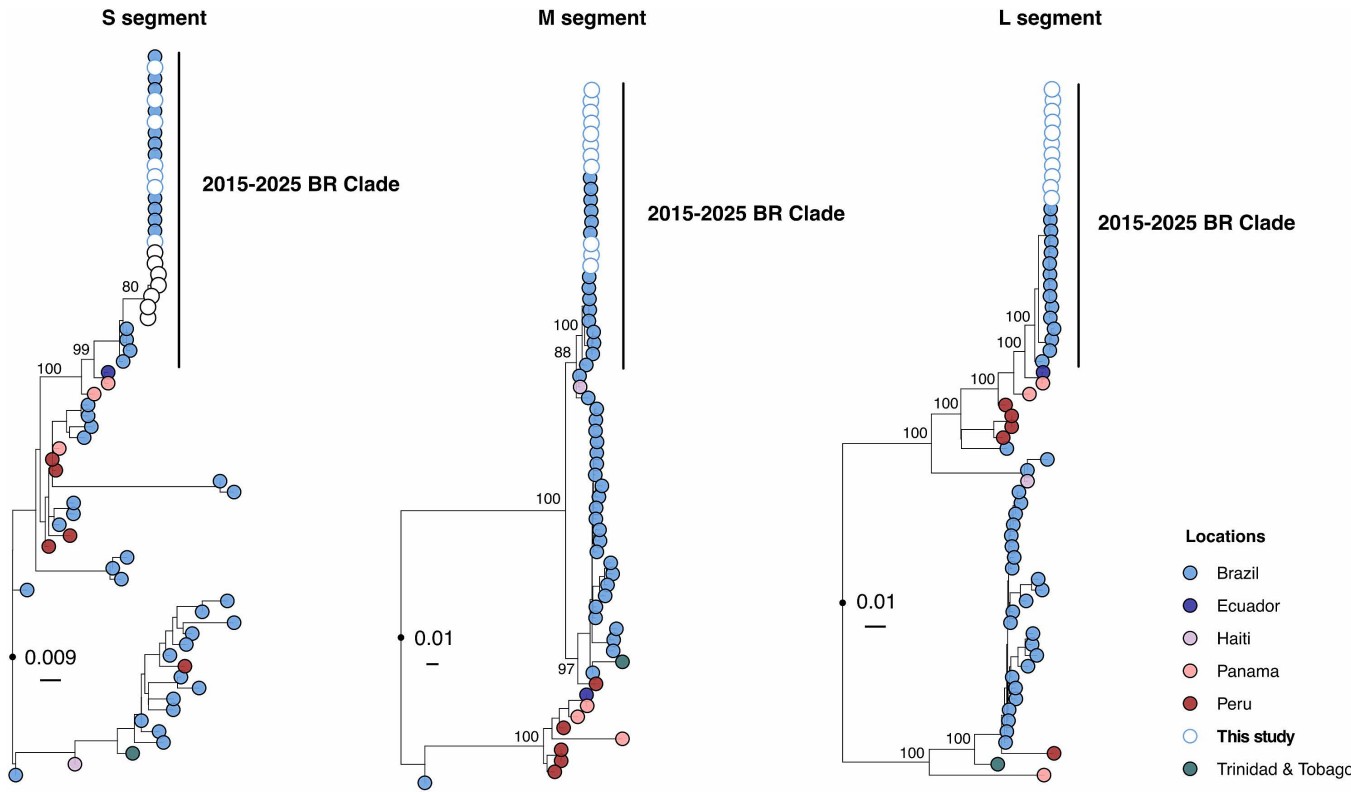

**Fig 3. Phylogenetic analysis of the OROV positive samples from Manaus, AM, Brazil, 2024.** Maximum likelihood phylogenetic tree of 64 representative OROV genomes, including twelve new genomes from this study. Phylogenetic trees are shown for the L segment (left), M segment (center), and S segment (right). The nodes are colored according to the location of each sample, specified in the legend. Phylogenies were midpoint rooted for clarity of presentation. Scale bar indicates the evolutionary distance of substitutions per nucleotide site. Bootstrap values based on 1,000 replicates are shown on principal nodes.

inflammatory (cytokine/chemokines) response affects melanocyte immune and metabolic functions contributing to development of skin rash like those presented in our patient groups [36,37]. Taken together, these results illustrate the clinical aspects associated with neurological, vascular, and immunological events during the OROV and DENV infections. The intensity, pattern, and rate of emergence of headaches and rash may potentially offer guidance in differential clinical diagnosis during co-circulation of the two infections.

Rash associated with OROV infection varies across studies, with reported rates ranging from 26.5% to 42% [38]. With cutaneous manifestations observed in 35% of our OROV patients during the initial evaluation, this is within the reported rates. A maculopapular rash diffusely distributed across the head, torso, and upper and lower limbs was the most common. These skin eruptions typically resolved spontaneously within a few days. Such variability in rash presentation may reflect differences in study populations, case definitions, or the timing of clinical assessments. Although the characteristics and duration of the rash are consistent with previous descriptions [39], additional features were observed during follow-up. Alongside fever and severe headache, the patients presented with a persistent diffuse rash and petechiae.

Both OROV and DENV infections present highly similar inflammatory pathways, with an overlap of the mediators both cases. However, OROV can induce a distinct immune response [40,41].

In both infections, CXCL10 and the regulatory cytokines IL-1RA and IL-10 were elevated at acute phase but diminishing towards recovery phase. With regards to DENV infection, our findings concur with a previous study reporting

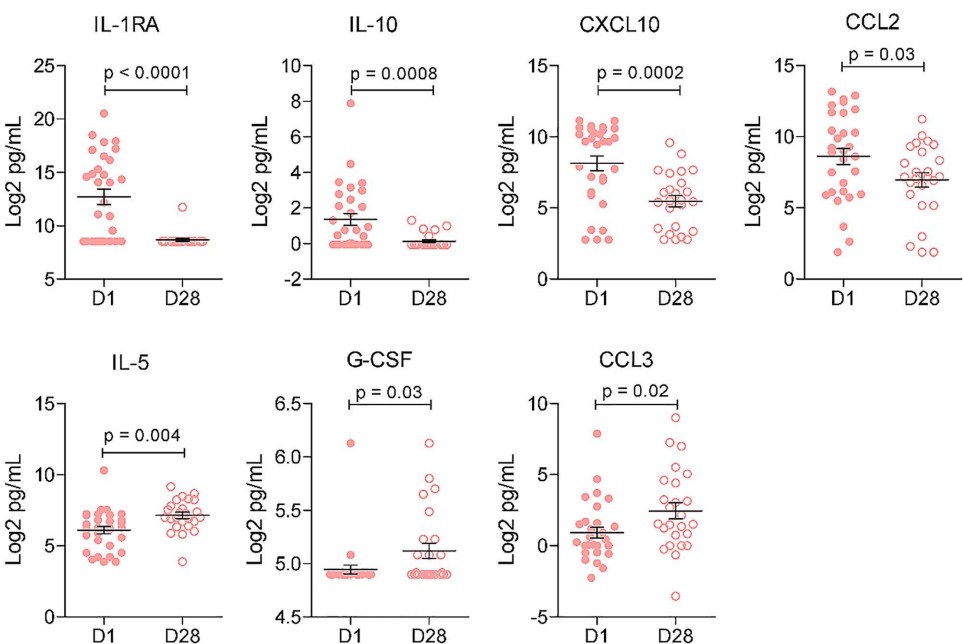

**Fig 4. Differential cytokine profile in the dengue virus infected patients at D1 and D28.**

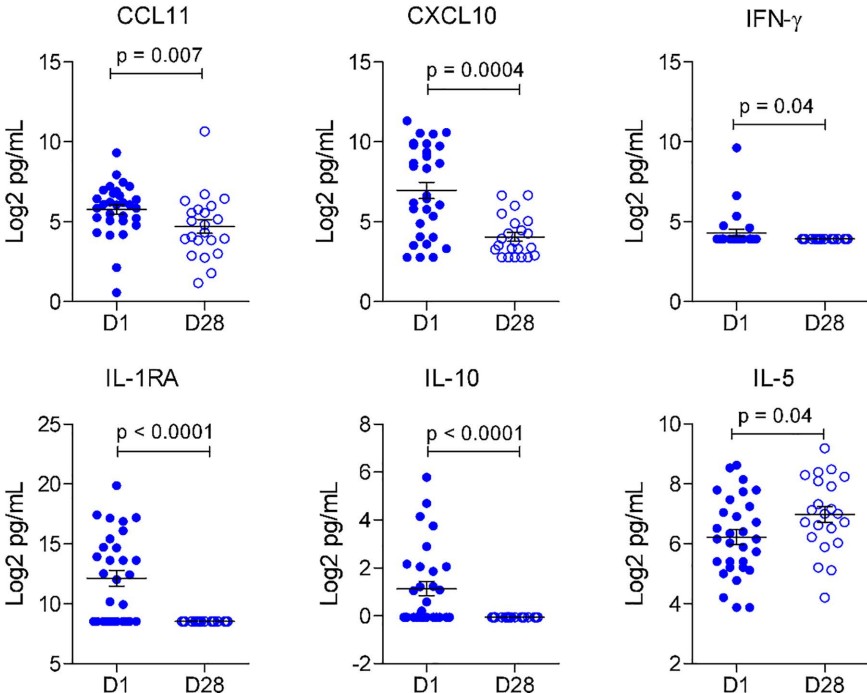

**Fig 5. Differential cytokine profile in the Oropouche virus infected patients at D1 and D28.**

that circulating CXCL10 and IL-1RA tend to be significantly associated with this infection [42]. As for OROV infection, authors have already reported a trend of increased IL-10 in the acute phase of infection, and a significant increase in CXCL10 in the first 3 days among late seroconverts patients (IgM/IgG negative at baseline), in addition to highlighting their association with innate immune events that could support type I IFN responses [40]. Interestingly, IL-5 levels were found lower in acute phase compared to recovery phase in both ORVOV and DENV infections. Higher levels of IL-5 would be anticipated after recovery since this could be related to the function of IL-5 in triggering B cell activation and differentiation into antibody-producing plasma cells [40,43]. Oliveira et al. (2019) reported a significant increase in IL-5 in late seroconverts with more than 11 days of OROV infection, converging with our observations of an increase towards the recovery phase [40]. However, differences of inflammatory mediators' profile were also observed. CCL2 was found to be elevated in DENV's but not OROV's acute phase. This cytokine is associated with monocyte recruitment, that is very important for viral clearance [42]. Increased levels of CCL2 in DENV infection was also reported in previous studies, even described as a potential prognostic biomarker that discern patients with higher risk of developing severe infection [42,44]. In OROV patients, CCL11 and IFN-γ were found increased in early phase of manifestations. To our knowledge, this is the first study to report the quantification of CCL11 in arboviral infection, which is associated to eosinophil, mast cells, basophils, neutrophils and macrophages chemotaxis [45]. Additionally CCL11 has a role in neuroinflammation, being able to induce neuronal cytotoxicity effects by stimulating the production of reactive oxygen species (ROS) in microglial cells [46]. These further hints at a potential relationship between CCL11 levels and neurological symptoms. It has been reported that regardless of the immunoglobulin profile, OROV fever patients presented higher levels of IFN-α at baseline as compared to uninfected donors [40]. Our study noted increased IFN-γ levels at acute phase in OROV cases contrary to the elevated IFN-γ among late seroconverted patients as reported by Oliveira et al. [40]. Our data bolsters the importance of interferon (IFN) in OROV infections, considering the importance of IFN in antiviral immunity and as a potent activator of macrophages [47]. Components of the type I IFN induction pathway, particularly the regulatory transcription factors IRF-3 and IRF-7, were demonstrated to have protective roles during OROV infection, though other transcriptional factors associated with an IFN response may also contribute to antiviral immunity against OROV [48,49]. G-CSF and CCL3 are known mediators involved in DENV infection [50,51]. In our data both were significantly elevated in the recovery phase, but only in the DENV cases. Taken with other factors, the profiles of inflammatory cytokines and chemokines provide valuable insights for monitoring these arboviral diseases. Furthermore, quantifying these markers enable analysis of their associations with other pathophysiological features.

Another relevant observation in this investigation addresses the increased levels of alanine transferase (ALT) in patients with OROV compared to DENV, even though no hepatitis was reported. Our findings agree with reported alterations in liver function indicators in both human and animal models of OROV infection [48,52,53]. However, in our study the ALT increases were mild, remaining within or slightly above normal limits, and were not clinically significant, likely due to the self-limiting nature of OROV infection. This observation may change in the event of severe OROV cases requiring hospitalization. The moderate increase in ALT observed reflects inflammation, but not to a degree indicative of severe liver damage or acute liver disease, considering the pivotal role hepatocytes play a in liver inflammatory response [54]. The findings here align with animal models of OROV infection, where elevated liver enzymes are indicative of liver inflammation [55–57].

One limitation of our study is that it only dealt with non-severe OROV cases, limiting our ability to assess clinical, laboratory and cytokine profiles associated with severity. A key strength, however, is that clinical characterization was done by physicians blinded to the patient's diagnostic status, minimizing the potential for observational bias. Another positive point is that in this work we included patients in a consecutive sampling, based on the free demand of health units that treat people with acute febrile syndrome, which comprehensively revealed the most prevalent clinical presentation of the cases.

## Conclusions

In a pioneering move, we described the clinical profile of OROV disease in a sample of patients from free-demand health-care units that treat people with acute febrile syndrome, which revealed the most prevalent clinical presentation of the cases, combined with laboratory and immunological aspects. The clinical overlap between OROV and DENV diseases highlights the critical need for improved differential diagnostics in endemic regions. While etiological diagnosis remains essential for epidemiological surveillance and outbreak control, it is equally important to recognize that, from a clinical standpoint, patient management, particularly in severe cases, should not be delayed by the pursuit of a specific diagnosis. This underscores the relevance of adopting an integrated approach to the management of acute febrile illness, especially in settings with co-circulation of multiple arboviruses. Diagnostic strategies should therefore prioritize tools capable of predicting disease severity and aiding in clinical decision-making, rather than focusing exclusively on pathogen identification. Our findings suggest that differential cytokine profiles, headache intensity, and rash presentation patterns hold potential as diagnostic indicators and should be further developed as adjunct tools in OROV and DENV diagnostics. Moreover, continuous genomic surveillance and broader clinical investigations are essential to track arbovirus evolution and transmission dynamics, including the introduction and spread of viral lineages, as well as the identification of potentially involved vectors and reservoirs. These efforts are particularly relevant in the context of viral reassortment, which may enhance both virulence and transmissibility. Investigating long-term neurological outcomes is also warranted. Given the geographic expansion of OROV, targeted public health interventions are urgently needed to mitigate future outbreaks and deepen understanding of its clinical and virological behavior.

## Supporting information

**S1 File. Primers and probes for Oropouche virus (OROV) detection using RT-qPCR.**
(DOCX)

**S2 File. Primers used for Oropouche virus amplicon-based next-generation sequencing.**
(DOCX)

**S3 File. Oropouche virus sequencing data information.**
(DOCX)

**S4 File. Oropouche virus genome sequences generated in this study and their respective accession numbers.**
The Oropouche virus sequencing data generated during the current study are available in the GISAID repository (GISAID. org/EpiArbo). To access and view the sequences, one needs to register to the site (https://gisaid.org/register/), then LOGIN to the GISAID repository. Once you have a login, you'll enter this area Epiarbo>Oropouche, and then click on search. You will then type in the sequence ID, of which it will appear. When you click on it (the ID), you can see the file information and download the sequences. In the case of OROV, it is an access number for the 3 segments.
(DOCX)

## Acknowledgments

We acknowledge the dedication and work put in by the support staff and research team in the REVISA project at the FMT-HVD hospital and UPA Campos Sales. We are also deeply grateful for the trust and cooperating of the recruited patients who made this possible.

## Author contributions

**Conceptualization:** Maria Paula Gomes Mourão, Gisely Cardoso de Melo, Flor Ernestina Martínez Espinosa, Vanderson Sampaio, Felipe Murta, Maurício Lacerda Nogueira, Marcus Lacerda, Wuelton Monteiro.

**Data curation:** Maria Paula Gomes Mourão, Victor Irungu Mwangi, Livia Sacchetto, Luiz Gustavo Gardinassi, Rebeca Linhares Abreu Netto, Jady Mota, Mayara Tavares, Liz Moreira Cordeiro, Dyandra dos Santos Porto, Bianca Daniele Silva de Oliveira, Carlos Eduardo Morais, Marco Aurélio Sartim, Wuelton Monteiro.

**Formal analysis:** Maria Paula Gomes Mourão, Gisely Cardoso de Melo, Joabi Nascimento, Victor Irungu Mwangi, Livia Sacchetto, Luiz Gustavo Gardinassi, Rebeca Linhares Abreu Netto, Jady Mota, Sergio Damasceno Pinto, Mayara Tavares, Barbara Seffair de Castro de Abreu Brasil, Bianca Daniele Silva de Oliveira, Ana Carolina Shuan Laco, Lorenna Pereira de Souza, Karolaine Oliveira Bentes, Flor Ernestina Martínez Espinosa, Vanderson Sampaio, Marco Aurélio Sartim, Maurício Lacerda Nogueira, Marcus Lacerda, Wuelton Monteiro.

**Funding acquisition:** Maria Paula Gomes Mourão, Gisely Cardoso de Melo, Vanderson Sampaio, Maurício Lacerda Nogueira, Marcus Lacerda, Wuelton Monteiro.

**Investigation:** Maria Paula Gomes Mourão, Gisely Cardoso de Melo, Joabi Nascimento, Victor Irungu Mwangi, Livia Sacchetto, Rebeca Linhares Abreu Netto, Sergio Damasceno Pinto, Mayara Tavares, Liz Moreira Cordeiro, Barbara Seffair de Castro de Abreu Brasil, Dyandra dos Santos Porto, Bianca Daniele Silva de Oliveira, Ana Carolina Shuan Laco, Lorenna Pereira de Souza, Karolaine Oliveira Bentes, Marcia Castilho, Carlos Eduardo Morais, Heline Silva Lira, Luís Felipe Alho, Felipe Murta, Marco Aurélio Sartim, Maurício Lacerda Nogueira.

**Methodology:** Maria Paula Gomes Mourão, Joabi Nascimento, Victor Irungu Mwangi, Livia Sacchetto, Luiz Gustavo Gardinassi, Rebeca Linhares Abreu Netto, Jady Mota, Bianca Daniele Silva de Oliveira, Lorenna Pereira de Souza, Karolaine Oliveira Bentes, Marcia Castilho, Carlos Eduardo Morais, Luís Felipe Alho, Vanderson Sampaio, Felipe Murta, Marco Aurélio Sartim, Marcus Lacerda.

**Project administration:** Gisely Cardoso de Melo, Joabi Nascimento, Heline Silva Lira.

**Resources:** Marcus Lacerda, Wuelton Monteiro.

**Supervision:** Maria Paula Gomes Mourão, Gisely Cardoso de Melo, Joabi Nascimento, Lorenna Pereira de Souza, Flor Ernestina Martínez Espinosa, Vanderson Sampaio, Marco Aurélio Sartim, Maurício Lacerda Nogueira, Marcus Lacerda, Wuelton Monteiro.

**Validation:** Maria Paula Gomes Mourão, Gisely Cardoso de Melo, Joabi Nascimento, Victor Irungu Mwangi, Livia Sacchetto, Luiz Gustavo Gardinassi, Rebeca Linhares Abreu Netto, Jady Mota, Sergio Damasceno Pinto, Marcia Castilho, Heline Silva Lira, Flor Ernestina Martínez Espinosa, Vanderson Sampaio, Felipe Murta, Maurício Lacerda Nogueira, Marcus Lacerda, Wuelton Monteiro.

**Visualization:** Victor Irungu Mwangi, Livia Sacchetto, Rebeca Linhares Abreu Netto, Jady Mota, Sergio Damasceno Pinto, Mayara Tavares, Liz Moreira Cordeiro, Barbara Seffair de Castro de Abreu Brasil, Dyandra dos Santos Porto, Bianca Daniele Silva de Oliveira, Ana Carolina Shuan Laco, Lorenna Pereira de Souza, Karolaine Oliveira Bentes, Marcia Castilho, Carlos Eduardo Morais, Heline Silva Lira, Luís Felipe Alho, Flor Ernestina Martínez Espinosa, Vanderson Sampaio, Felipe Murta, Marco Aurélio Sartim, Maurício Lacerda Nogueira, Wuelton Monteiro.

**Writing – original draft:** Maria Paula Gomes Mourão, Gisely Cardoso de Melo, Joabi Nascimento, Victor Irungu Mwangi, Luiz Gustavo Gardinassi, Bianca Daniele Silva de Oliveira, Marco Aurélio Sartim, Maurício Lacerda Nogueira, Wuelton Monteiro.

**Writing – review & editing:** Maria Paula Gomes Mourão, Gisely Cardoso de Melo, Joabi Nascimento, Victor Irungu Mwangi, Livia Sacchetto, Luiz Gustavo Gardinassi, Rebeca Linhares Abreu Netto, Jady Mota, Sergio Damasceno Pinto, Mayara Tavares, Liz Moreira Cordeiro, Barbara Seffair de Castro de Abreu Brasil, Dyandra dos Santos Porto, Bianca Daniele Silva de Oliveira, Ana Carolina Shuan Laco, Lorenna Pereira de Souza, Karolaine Oliveira Bentes, Marcia Castilho, Carlos Eduardo Morais, Heline Silva Lira, Luís Felipe Alho, Flor Ernestina Martínez Espinosa, Vanderson Sampaio, Felipe Murta, Marco Aurélio Sartim, Maurício Lacerda Nogueira, Marcus Lacerda, Wuelton Monteiro.

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
