## [Decision Letter · Decision Letter 0]

22 Jun 2025

PNTD-D-25-00611

Clinical and Laboratory Profiles of Oropouche Virus Disease and Dengue from 2024 outbreak in Manaus, Brazilian Amazon

Dear Dr. Monteiro,

Gomes Mourão et al submitted to the PLOS Neglected Tropical Diseases the manuscript entitled “Clinical and Laboratory Profiles of Oropouche Virus Disease and Dengue from 2024 outbreak in Manaus, Brazilian Amazon.” A novel reassortant of Oropouche virus (OROV) is causing the largest outbreak in history, with over 16,000 cases confirmed by RT-qPCR and, for the first time fatalities, vertical transmission, and Guillain-Barré. Furthermore, the spread of the virus to non-endemic Latin American countries and the number of exported cases to the US and Europe have been unexpected. The pathophysiology of the OROV disease is not completely understood.

Gomes Mourão et al research contributes to the better understanding of the OROV disease. In this work, clinical findings and cytokine profiles of Dengue patients are compared to OROV patients. Readers would find interesting the cytokine profile of OROV patients. While both viruses, Oropouche and Dengue, cause disease, in the title the OROV disease is highlighted but not the Dengue disease.

Thank you for submitting your manuscript to PLOS Neglected Tropical Diseases. After careful consideration, we feel that it has merit but does not fully meet PLOS Neglected Tropical Diseases's publication criteria as it currently stands. Therefore, we invite you to submit a revised version (Major Revision) of the manuscript that addresses the points raised during the review process. Special attention should be paid to the comments of reviewers 2 and 3.

Please submit your revised manuscript within 60 days Aug 21 2025 11:59PM. If you will need more time than this to complete your revisions, please reply to this message or contact the journal office at plosntds@plos.org. Please include the following items when submitting your revised manuscript:

We look forward to receiving your revised manuscript.

Kind regards,

Daniel Limonta, MD, PhD

Academic Editor

Mabel Carabali

Section Editor

Shaden Kamhawi

co-Editor-in-Chief

Paul Brindley

co-Editor-in-Chief

**Journal Requirements:**

1) Please provide an Author Summary. This should appear in your manuscript between the Abstract (if applicable) and the Introduction, and should be 150-200 words long. The aim should be to make your findings accessible to a wide audience that includes both scientists and non-scientists. Sample summaries can be found on our website under Submission Guidelines:

2) We noticed that you used the phrase 'data not shown' in the manuscript. We do not allow these references, as the PLOS data access policy requires that all data be either published with the manuscript or made available in a publicly accessible database. Please amend the supplementary material to include the referenced data or remove the references.

4) Tables should not be uploaded as individual files. Please remove these files and include the Tables in your manuscript file as editable, cell-based objects. For more information about how to format tables, see our guidelines:

https://journals.plos.org/plosntds/s/tables 

5) We have noticed that you have uploaded Supporting Information files, but you have not included a list of legends. Please add a full list of legends for your Supporting Information files after the references list.

6) In the online submission form, you indicated that "The analyzed data sets generated during the study are available from the corresponding author on reasonable request." All PLOS journals now require all data underlying the findings described in their manuscript to be freely available to other researchers, either

1. In a public repository

2. Within the manuscript itself

3. Uploaded as supplementary information.

7) Please amend your detailed Financial Disclosure statement. This is published with the article. It must therefore be completed in full sentences and contain the exact wording you wish to be published.

3) If any authors received a salary from any of your funders, please state which authors and which funders.

8) Please ensure that the funders and grant numbers match between the Financial Disclosure field and the Funding Information tab in your submission form. Note that the funders must be provided in the same order in both places as well. Currently, "University of Texas Medical Branch" is missing from the Funding Information tab.

**Comments to the Authors:**

**Please note that one of the reviews is uploaded as an attachment.**

**Reviewers' Comments:**

Reviewer's Responses to Questions

**Key Review Criteria Required for Acceptance?**

**Methods**

-Are the objectives of the study clearly articulated with a clear testable hypothesis stated?

-Is the study design appropriate to address the stated objectives?

-Is the population clearly described and appropriate for the hypothesis being tested?

-Is the sample size sufficient to ensure adequate power to address the hypothesis being tested?

-Were correct statistical analysis used to support conclusions?

-Are there concerns about ethical or regulatory requirements being met?

Reviewer #1: (No Response)

Reviewer #2: Methods

Study design

Line 128 – Why the study excluded the children under 5 years? Clinical differences may exist according to age, that should be considered, evaluated and reported to the community. If ethical or other concerns exist, please clarify.

Line 133-139 – Please make the symptoms list more concise, referring to table or results.

Line 146 – Why molecular analysis was limited to OROV and DENV, considering the previously mentioned arboviruses (i.e.: CJKV. ZIKV…)? They are also mentioned in line 156 in molecular assays section.

Laboratory diagnosis of OROV and dengue

Line 150 – Is the OROV RT-PCR a real quantitative assay, thus implying the use of calibration curve and standards?

Line 151 – PRNT is a test for the detection of antibodies, not for determination of viral presence or titration. How authors used it? No result is reported in the work.

Line 155-158 – How DENV serotype was determined? Why samples (or patients) positive for CHKV and ZIKV were excluded?

Virus genotyping

Line 164-171 – Was the S RT-PCR different from the one used for OROV infection diagnosis? Why was a second RT-PCR performed, if a data was already available from first one? In addition, to be uniform the molecular analyzer should be reported also in the previous section.

Line 169 – The Ct value is not a mean of quantification, if anything a semi-quantification one. Quantification always need a standard curve to be performed, as mentioned before.

Line 173-174 – Please rewrite the sentence: it is not very clear how the protocol and kit were adapted for OROV sequencing.

Neurological markers and Cytokines and chemokines

The methods used for neurological and inflammatory markers are the same: please unify the paragraphs or synthesize them, including a reason for selecting the reported markets.

Reviewer #3: The network of symptoms analysis requires more explanation. What was the question or hypothesis being tested? Why were you looking at the numbers of patients sharing two symptoms? Is this similar to something like a cluster or latent class analysis? Explanation might require supplemental material.

The sequencing methods should be reviewed by a laboratorian familiar with these methods.

Specific comments: several symptoms require clarification and might be related to translation inaccuracy (lines 136-139): "adenomegaly" - should this be lymphadenopathy?; "conjunctival congestion" - conjunctival injection? (this is also duplicated so one should be deleted); "oropharyngeal changes" - what does this mean?

**Results**

-Does the analysis presented match the analysis plan?

-Are the results clearly and completely presented?

-Are the figures (Tables, Images) of sufficient quality for clarity?

Reviewer #1: (No Response)

Reviewer #2: Results

Line 239 – What does it mean “by a consecutive sampling”? The fact that samples were collected at D1 and D28?

Line 241 – Check the sentence: there is probably a duplication of verb.

Line 243-248 – The paragraph is not very clear. Authors should make they results more readable. For example, there is the repetition of “on the admission”, as well as the symptoms associated with OROV infection are dispersed in across sentences, making difficult to understand which are really important compared to DENV positive population.

Line 281-282 – Firstly, the verb “illustrated” is not proper for describing the outcome of clinically relevant assays: please change it. Most importantly, despite difference in ALT concentration between OROV and DENV patients, the mean (or median?) level is in the normality range: this is an important aspect to consider, because the added value of such result is to support a possible virus-driven hepatic damage.

Line 295-302 – Try to reduce data on sequencing quality to very essential ones; in addition, figure 3 is not informative, with no added value on the results. Focusing on phylogenesis is recommended.

In figure 4 is not clear which the study sequences among.

Line 325-331 – Please rewrite the paragraph: the sentences are redundant, with the same scheme and words repeated. It should be more readable and harmonized.

Figure 5 is not informative.

Line 357-441 – There are a lot of data presented in this section. Even interesting, the presentation is not very clear. Please try to underline most important outcomes, to let readers to understand what are differences between DENV and OROV infections: such non-viral parameters could drive physicians in differential diagnosis. In addition: there was an equal distribution between the S1, S2, S3 groups to have reliable data on neurological markers? Even p-values supported data, a reduced cohort could limit the relevance.

Figure 8 G-H is not very clear. Are they correlations between a non-viral parameter and headache intensity? How can be deduced the adjustment for age and gender?

Reviewer #3: The figures and tables should all be placed at the end of the manuscript and the titles and captions should be placed with the appropriate figures and tables. It's very hard to follow. Also the figures are very low resolution and are hard to read.

Lines 394-430 are impossible to understand - there is too much detail, terminology, and disconnected correlations. I don't know what any of it means. A single table or figure demonstrating the relevant correlations noted for OROV and DENV with the text highlighting the notable findings would be much more understandable.

There are places in the results where text should be in the methods or discussion - e.g., paragraph from lines 376-380: first sentence is methods; paragraph from lines 432-441: first sentence is methods; last sentence is discussion.

**Conclusions**

-Are the conclusions supported by the data presented?

-Are the limitations of analysis clearly described?

-Do the authors discuss how these data can be helpful to advance our understanding of the topic under study?

-Is public health relevance addressed?

Reviewer #1: (No Response)

Reviewer #2: Discussion

Some part of the discussion are a repetition of results: please change and adapt to a new form.

Review the discussion according to changes in the above sections.

Reviewer #3: It's unclear whether the conclusions are supported by the data presented because the results are not presented in an understandable way as mentioned above.

The sequencing data seem like a distinct topic from the rest of the paper. Can the authors connect this information to the clinical findings? Otherwise, I would recommend shortening the sequencing methods (or put in a supplemental section), results, and discussion.

Would not present new data in the discussion (line 488).

There is little discussion about the differences in symptoms (other than headache) between Oropouche virus disease and dengue.

Overall, the discussion needs to be streamlined and organized by topic as discussed below.

**Editorial and Data Presentation Modifications?**

Reviewer #1: (No Response)

Reviewer #2: (No Response)

Reviewer #3: Much of the lack of clarity could be related to language translation errors; the paper should be reviewed for correct translation to English. For example, in the intro (line 95) Culicoides should be biting midges rather than mosquitoes.

**Summary and General Comments**

Reviewer #1: (No Response)

Reviewer #2: The paper by Maria Paula Gomes Mourão et al. data from a study on Oropouche virus and Dengue virus infections in Brazil. Authors described the clinical course of included subjects, also measuring blood, biochemical and neurological markers. Interestingly, some specific association between viruses, symptoms and biomarkers were found.

The work well written, even revision on puncture, verbs and sentences construction should be performed. In addition, some passages are redundant and confounding, needing revision.

Reviewer #3: This is an incredibly rich dataset that has great potential to provide useful information on the longitudinal clinical, laboratory, and pathophysiology of Oropouche virus disease vs. dengue. However, the paper is so long and confusing and relatively disorganized that it is hard to draw any conclusions from the data. There are multiple questions that could be addressed with these data (perhaps in more than one manuscripts) on 1) the pathophysiology of disease (e.g., pathogenesis of headache and other symptoms as it relates to evolution of cytokine responses and neurological biomarkers, CNS neuroinvasion vs. inflammatory-mediated or vascular changes), 2) differences in clinical and biomarker profiles between Oropouche virus disease and dengue - e.g., are there clinical profiles that are predictive of one or the other or of severe outcomes? 3) are there genetic mutations associated with clinical syndromes or severity of disease? Overall, the authors need to clarify their hypotheses and organize the methods, results, and discussion accordingly.

Also, the paper needs to be streamlined and shortened considerably.

The abstract doesn't make it clear which symptoms, cytokines, and biomarkers are different between Oropouche virus disease and dengue virus.

PLOS authors have the option to publish the peer review history of their article (what does this mean? ). If published, this will include your full peer review and any attached files.

**Do you want your identity to be public for this peer review?** For information about this choice, including consent withdrawal, please see our Privacy Policy .

Reviewer #1: No

Reviewer #2: No

Reviewer #3: No

**Figure resubmission:**

**Reproducibility:**



---

## [Editor Report · Decision Letter 1]

25 Sep 2025

Dear Dr. Monteiro,

We are pleased to inform you that your manuscript 'Clinical and Laboratory Profiles of Oropouche Virus Disease from the 2024 Outbreak in Manaus, Brazilian Amazon' has been provisionally accepted for publication in PLOS Neglected Tropical Diseases. The immunopathogenesis of Oropouche virus remains underexplored. With the recent emergence and rapid spread in the Americas of a novel reassortant strain from the Brazilian Amazon, there is a pressing need for further clinical and immunological investigations. Gomes Mourão et al. present a comparative analysis of symptomatology and clinical parameters in patients with Dengue and Oropouche virus infections, alongside data on circulating immunological markers. The study offers valuable insights into the clinical presentation and immune response associated with Oropouche virus.<o:p></o:p>

The authors have undertaken a thorough revision of their manuscript, addressing several key concerns raised during peer review. One of the most significant points highlighted by reviewers was the need for improved clarity and precision in the manuscript’s wording. Additionally, Figure 3’s background color was noted as a potential issue. Although the figure was updated in the revised submission, our editorial team will work with the authors to ensure the final version meets publication standards.<o:p></o:p> Before your manuscript can be formally accepted you will need to complete some formatting changes, which you will receive in a follow up email. A member of our team will be in touch with a set of requests.

Best regards,

Daniel Limonta, MD, PhD

Academic Editor

Mabel Carabali

Section Editor

Shaden Kamhawi

co-Editor-in-Chief

Paul Brindley

co-Editor-in-Chief

---

## [Editor Report · Acceptance letter]

Dear Dr. Monteiro,

We are delighted to inform you that your manuscript, " 

Clinical and Laboratory Profiles of Oropouche Virus Disease from the 2024 Outbreak in Manaus, Brazilian Amazon," has been formally accepted for publication in PLOS Neglected Tropical Diseases.

Best regards,

Shaden Kamhawi

co-Editor-in-Chief

Paul Brindley

co-Editor-in-Chief
